# BecomingLit: Relightable Gaussian Avatars with Hybrid Neural Shading

**Jonathan Schmidt**     **Simon Giebenhain**     **Matthias Nießner**
Technical University of Munich

`jonathsch.github.io/becominglit`

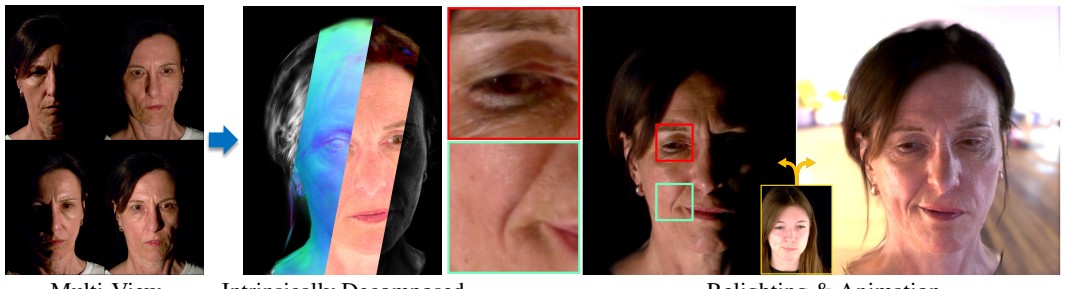

Multi-View OLAT Dataset     Intrinsically Decomposed Avatar     Relighting & Animation

Figure 1: **BecomingLit**: Our approach effectively reconstructs detailed human head avatars that can be animated from videos and relighted in real-time using our hybrid neural shading approach. Besides our method, we introduce a new high-quality, multi-view OLAT dataset of faces.

## Abstract

We introduce *BecomingLit*, a novel method for reconstructing relightable, high-resolution head avatars that can be rendered from novel viewpoints at interactive rates. Therefore, we propose a new low-cost light stage capture setup, tailored specifically towards capturing faces. Using this setup, we collect a novel dataset consisting of diverse multi-view sequences of numerous subjects under varying illumination conditions and facial expressions. By leveraging our new dataset, we introduce a new relightable avatar representation based on 3D Gaussian primitives that we animate with a parametric head model and an expression-dependent dynamics module. We propose a new hybrid neural shading approach, combining a neural diffuse BRDF with an analytical specular term. Our method reconstructs disentangled materials from our dynamic light stage recordings and enables all-frequency relighting of our avatars with both point lights and environment maps. In addition, our avatars can easily be animated and controlled from monocular videos. We validate our approach in extensive experiments on our dataset, where we consistently outperform existing state-of-the-art methods in relighting and reenactment by a significant margin.

## 1 Introduction

The creation of photorealistic, relightable 3D head avatars from real-world data is a core problem of computer vision with applications across a wide range of graphics tasks, such as cinematography, virtual reality, or the metaverse in general. Traditionally, this requires professional, room-scale capture setups that only a handful of institutions can afford [34, 4, 23, 24], as the joint estimation of geometry, intrinsic material parameters, and lighting is an extremely under-constrained problem.

39th Conference on Neural Information Processing Systems (NeurIPS 2025).

At the same time, with the progressing growth of virtual reality applications at the consumer level, creating photorealistic avatars is becoming more important than ever. While there has been immense progress over the recent years in terms of geometric representations [15, 30, 10, 18, 25], visual quality and rendering speed thanks to the availability of custom datasets [16], most 3D avatars do not have a disentangled representation of the material properties and bake the radiance properties of the training environment into the avatar, which makes relighting impossible. As a result, placing the avatar in a novel virtual environment dramatically lowers the visual quality of the renderings. In comparison, research on relightable avatars is scarce. One of the major reasons for this is the lack of publicly available and free-to-use datasets that come with controlled light captures in order to broadly study the reconstruction of facial appearance.

To this end, we introduce an OLAT dataset and propose *BecomingLit*, a novel approach to reconstruct photorealistic, relightable head avatars from short multi-view light stage sequences. We represent the head with expression-dependent Gaussian primitives and model the complex reflection behavior of faces by learning a hybrid neural BRDF. Thanks to our efficient parameterization and regularization, our method requires a capture setup that is an order of magnitude more economical compared to previous work, and outperforms state-of-the-art methods in self-reenactment under novel illuminations. To address the lack of data, we introduce a new multi-view video dataset of different participants in a light stage setting, which we will make publicly available for research purposes. Overall, our contributions are two-fold:

- We introduce a novel, publicly available dataset, combining high-resolution, high-framerate, multi-view recordings of different subjects in a calibrated light stage setting.

- We propose a relightable, photorealistic avatar representation based on 3D Gaussian primitives and hybrid neural shading, which can be relighted and rendered from novel viewpoints in real-time and animated from monocular videos.

## 2 Related Work

**Human Head Modeling** addresses the problem of representing and modeling the geometry and appearance of human heads. Traditional methods learn morphable models from head scans via PCA [5, 28, 17]. While being strong in generalization, PCA-based 3DMMs have a limited expressiveness and can fail to represent fine geometric details such as skin wrinkles or hair. As an alternative, [4, 20] propose to learn the geometry and appearance space with autoencoders. More recently, volumetric approaches based on NeRF [25], represent heads with more detailed appearance, despite not requiring explicit input geometry [16, 54, 18, 57, 8, 32]. Another line of work uses 3D Gaussian primitives [15] to model human heads [10, 19, 47], some of them in combination with a 3D morphable model [30, 34].

**Facial Appearance Capture.** Capturing the appearance of human faces is a long-standing problem in computer vision. Debevec et al. [7] introduced the light stage and demonstrated how the reflectance field of a human face can be reconstructed from one-light-at-a-time captures, and relighted using image-based rendering [7, 43]. Subsequent work leveraged polarized light to decompose specular and diffuse reflectance [21, 9, 11, 33, 2]. [35, 46] approach the intrinsic decomposition problem with radiance fields [25]. [4, 49, 49] propose a learnable, data-driven appearance model that learns avatar relighting in an end-to-end manner with a neural lighting model. In constrast, [34, 40] propose to learn radiance transfer properties of 3D Gaussian primitives [15].

**Neural Shading** is concerned with learning light reflectance functions instead of using analytical models developed in computer graphics. This has been successfully applied to static scenes [35, 46, 48, 53, 31] and dynamic objects [23, 24]. Image-based methods enable relighting of a single portrait [12, 26, 38, 41, 51], but fail to synthesize novel views and struggle with temporal consistency, which are key requirements for head avatars. While [34] learns the coefficients of an explicit precomputed radiance transfer (PRT) function, [31] proposes to learn the PRT function with a neural network. In contrast, we propose a hybrid neural shading approach, combining implicitly learned diffuse radiance transfer with a well-established analytical specular term.

## 3 Multi-View OLAT Dataset of Faces

Capturing human faces under known, calibrated illumination enables efficient estimation of skin properties such as reflectance [7, 34, 35] and pore-level normals [21]. We therefore introduce a novel dataset, which consists of multi-view recordings of different subjects in a light stage setting.

Table 1: Existing light stage datasets of human heads. ICT-3DRFE [37] contains only processed data and no raw footage or calibration data.

| Dataset | # IDs | # Views | # Lights | FPS | Resolution | Setup Cost |
|---|---|---|---|---|---|---|
| 3DRFE [37] | 23 | | only processed data | | | $$ |
| Goliath [22] | 4 | 144 | 460 | 9 (90)[1] | 1334x2048 | $$$ |
| Ours | 10 | 16 | 40 | 72 | 2200x3208 | $ |

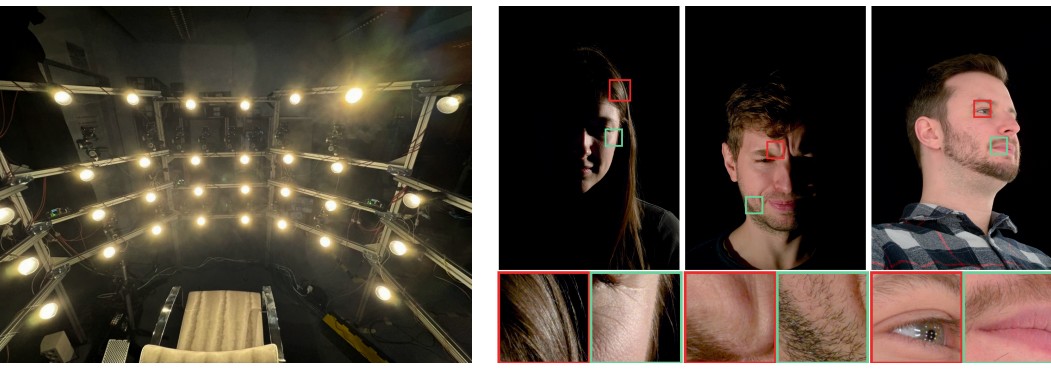

(a) Capture Rig  (b) Dataset Samples

Figure 2: **OLAT Dataset**: (a) Our custom light-stage rig we used to capture (b) our dataset consisting of high-resolution, high frame rate, multi-view recordings of faces under both OLAT and fully-lit conditions.

The dataset offers an unprecedented combination of high-resolution, high-frame-rate multi-view recordings of many sequences under numerous calibrated lighting conditions.

## 3.1  Capture Setup

As our primary capture target are human faces, we build a light stage setup that covers the frontal hemisphere of the subject's head. The setup consists of 16 machine vision cameras and 40 custom-built LED modules that are driven by microcontrollers. The cameras cover a field of view of 93° horizontally and 32° vertically. The lights are placed uniformly around the subject, covering a range of 180° horizontally and 60° vertically. All cameras and lights face towards the subject's face. See Figure 2 for a visualization of the capture rig. Each LED emits enough luminance to run both one-light-at-a-time (OLAT) and more complex light patterns, while maintaining a low shutter speed of 3ms, thus reducing motion blur to a minimum. We use high-quality LEDs with a Color Rendering Index (CRI) of over 98, which closely approximates natural white light.

We control the LEDs using microcontrollers that we synchronize with the cameras using a vendor-specific logic. Our capture rig is equipped with 16 machine vision cameras, which we internally synchronize using the Precision Time Protocol (PTP), leading to multi-view frames captured with a deviation of less than one microsecond. Each of the cameras records images with a resolution of 2,200x3,208 pixels at 72 frames per second, sufficient to capture specular reflections on the skin at pore-level detail as depicted in Figure 2b.

## 3.2  Data Acquisition

Using our light stage setup, we capture several sequences of different participants for a few minutes in total. During the capture sessions, each participant performs a predefined set of facial expressions, emotions, and reads out several sentences. Please refer to the supplementary for more details about our capture script. In total, we record around 150 seconds for each subject, which is divided into 6 blocks. In addition, we capture another sequence where every participant is free to perform arbitrary expressions for 20 seconds. For each frame, we activate a new light from the set of available OLAT configurations. To enable tracking, we follow previous work [43, 34] and interleave our cycle of light patterns with fully-lit tracking frames. More specifically, every third frame is a tracking frame, which

---

[1]90 FPS is only available for a short test segment.

results in tracking sequences captured at effectively 24 frames per second. See the rightmost image of Figure 2b for an example of a tracking frame.

### 3.3 Data Processing

The camera poses and intrinsic parameters are obtained using a checkerboard and bundle adjustment. Both the position and the intensity of the LEDs are calibrated using a mirror sphere whose shape and reflection properties are known. We follow the procedure of [45] and find the 3D position of each light source using ray-tracing in a multi-view capture of the mirror sphere. To account for differences in colors among the camera sensors, we use a color checker board and compute a color correction matrix for each camera. We use BiRefNet [55] for obtaining high-resolution foreground masks and obtain semantic segmentation with Facer [56].

### 3.4 Data Privacy

Our dataset contains highly personal information, which requires distributing it with extreme caution. We will only share the data with approved academic institutions and exclusively for non-commercial research purposes. All participants signed an agreement for publication, yet retain the right to have their data deleted at any time in the future, which we will enforce when distributing the dataset.

## 4 Method

Our method reconstructs relightable avatars from multi-view light stage sequences. Figure 3 provides an overview of our method. After preliminary information (Sec. 4.1), we describe our geometry (Sec. 4.2) and appearance (Sec. 4.3) model. In Sec. 4.4 and 4.5, we provide details about the optimization strategy and implementation details, respectively.

### 4.1 Preliminaries

**3D Gaussian Splatting** [15] introduces a point-based radiance field representation, that defines a 3D scene with a set of anisotropic 3D Gaussians parameterized by mean $\boldsymbol{\mu}$, covariance $\boldsymbol{\Sigma}$ and opacity $\sigma$. In addition, each Gaussian can hold an arbitrary number of features. Unlike continuous representations such as NeRF [25] that require ray marching for rendering, 3D Gaussians can be efficiently projected onto the image plane and rasterized in real-time on consumer-grade GPUs. We refer to the original paper of [15] for a more thorough overview.

**Physically-based Rendering** aims at synthesizing images by simulating the physical transport of light from the emitter to the camera sensor. The core is the rendering equation [13] that is defined as follows:

$$L_o(\boldsymbol{x}, \boldsymbol{\omega_o}) = \int_{\Omega} f_r(\boldsymbol{x}, \boldsymbol{\omega}_i, \boldsymbol{\omega}_o) L_i(\boldsymbol{x}, \boldsymbol{\omega}_i)(\boldsymbol{\omega}_i \cdot \boldsymbol{n}) d\boldsymbol{\omega}_i \qquad (1)$$

where $L_o$ is the outgoing radiance observed by the camera, $L_i$ is the incident radiance at point $\boldsymbol{x}$ from direction $\boldsymbol{\omega}_i$, and $f_r$ is the BRDF. Our goal is to recover the BRDF $f_r$ from data observations, such that the resulting avatar can be integrated with novel illuminations.

### 4.2 Geometry

We model the geometry of our avatar with a fixed set of anisotropic Gaussians [15] that we define on the UV map of a tracked template mesh. Inspired by [40, 34, 10], we employ an expression-dependent dynamics module $\mathcal{F}_g$, and a view and expression-dependent module $\mathcal{F}_v$ to model fine-grained geometric expression details beyond the scope of the template mesh.

As our base geometry, we use the parametric head model FLAME [17], which models coarse deformations over time. Given the fully-lit tracking frames of our dataset, we obtain shape, expression, and pose parameters using the photometric tracker VHAP [30, 29]. For the remaining OLAT frames, we linearly interpolate the FLAME parameters of the nearest tracking frames. To get the proxy geometry for the Gaussian primitives, we obtain the posed mesh $\mathcal{M} = (\mathcal{V}, \mathcal{F})$ from the FLAME parameters and compute tangents $\boldsymbol{t}_k$, bitangents $\boldsymbol{b}_k$ and normals $\boldsymbol{n}_k$ for every texel $k$ on the UV map. In addition, we obtain the interpolated 3D position of texel $k$, denoted as $\hat{\boldsymbol{\mu}}_k$.

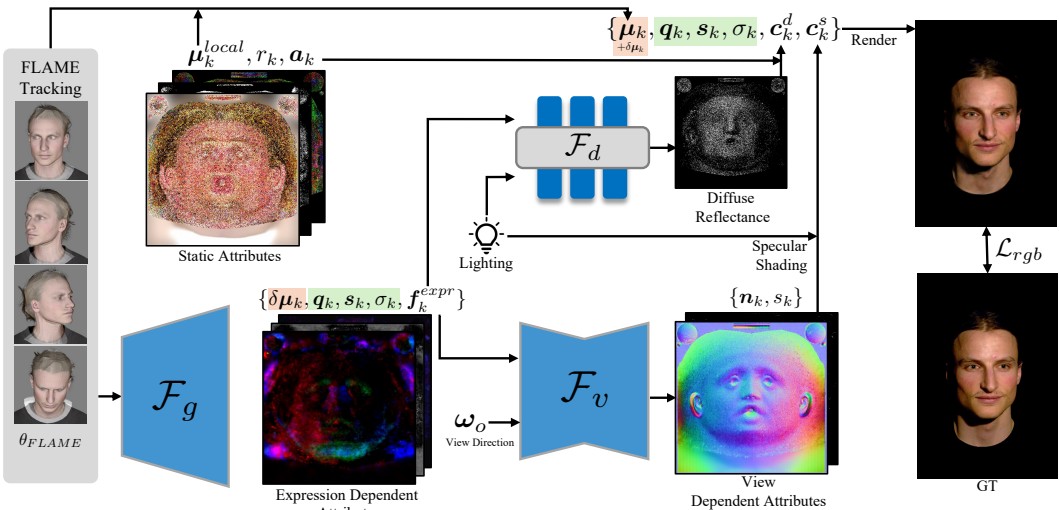

Figure 3: **Method Overview**: Given estimated FLAME coefficients, we obtain posed 3D Gaussian primitives with our expression-dependent dynamics module $\mathcal{F}_g$. To render photorealistic appearance, we combine the neural diffuse BRDF $\mathcal{F}_d$ with an analytical specular shading term. The parameters for the specular shading are predicted by the view-dependent $\mathcal{F}_v$ network. The avatar is optimized from light stage sequences using a photometric loss term.

Given the tracked FLAME expression parameters $\theta_{FLAME}$, we define $\mathcal{F}_g$ as a convolutional neural network which predicts per-gaussian attributes in UV-space:

$$\{\delta\boldsymbol{\mu}, \boldsymbol{q}, \boldsymbol{s}, \sigma, \boldsymbol{f}^{expr}\}_{k=1}^{M} = \mathcal{F}_g(\theta_{FLAME}) \tag{2}$$

The final Gaussian center $\boldsymbol{\mu}_k$ is defined as $\boldsymbol{\mu}_k = \hat{\boldsymbol{\mu}}_k + R_k^{TBN}\boldsymbol{\mu}_k^{local} + \delta\boldsymbol{\mu}$, where $R_k^{TBN} = [\boldsymbol{b}_k, \boldsymbol{t}_k, \boldsymbol{n}_k]$ is the orientation of the shading frame of texel $k$, and $\boldsymbol{\mu}_k^{local}$ is a parameter learned statically for each gaussian. The purpose of $\boldsymbol{\mu}_k^{local}$ is to define most of the offsets expression-independent, such that we can regularize the expression-dependent offsets $\delta\boldsymbol{\mu}_k$ to be small, which avoids artifacts when synthesizing novel expressions. The remaining Gaussian parameters $\boldsymbol{q}_k, \boldsymbol{s}_k, \sigma_k$ are directly predicted by $\mathcal{F}_g$. $\boldsymbol{f}_k^{expr}$ is an expression-dependent feature vector for shading, which we describe in Sec. 4.3.

### 4.3 Material

Modeling the reflectance properties of faces with common analytical models from computer graphics inevitably leads to insufficient quality due to their lack of modeling global illumination effects, such as subsurface scattering, which is omnipresent on human skin. We observe that such global illumination effects primarily affect the low-frequency, view-independent diffuse part. Thus, we propose a hybrid shading scheme, which learns diffuse light transport implicitly with a small neural network, while modeling specular reflectance with a well-established analytical model. To this end, we decompose the reflectance function $f_r(\boldsymbol{\omega}_o, \boldsymbol{\omega}_i)$ from Eq. (1) into a view-independent diffuse term $f_d(\boldsymbol{\omega}_i)$ and a view-dependent specular term $f_s(\boldsymbol{\omega}_o, \boldsymbol{\omega}_i)$.

**Diffuse.** The view-independent diffuse term models subsurface scattering and self-shadowing effects. At its core is a tiny neural network $\mathcal{F}_d$, shared among all primitives, and jointly trained with the avatar. The final diffuse color $\boldsymbol{c}_k^d$ is computed by multiplying statically learned albedo $\boldsymbol{a}_k$ with the predicted reflectance of $\mathcal{F}_d$:

$$\boldsymbol{c}_k^d = \boldsymbol{a}_k \, \mathcal{F}_d(SH_m(\boldsymbol{L}_i), \boldsymbol{f}_k^{expr}) \tag{3}$$

where $SH_m(\boldsymbol{L}_i)$ are the coefficients from the spherical harmonics parameterization of the incident light of degree $m$, and $\boldsymbol{f}_k^{expr}$ are the expression-dependent feature vectors. We empirically set SH degree $m$ to 6 in all experiments.

$\mathcal{F}_d$ is parameterized as a monochrome BRDF function, mapping single-channel incident light to a scalar reflectance value. This parameterization is necessary as the model only sees white light during

training, yet must also handle colored illumination at inference time. Here, we evaluate $\mathcal{F}_d$ separately for each color channel and concatenate the results into a single reflectance vector, which we multiply element-wise with the albedo. The architecture of $\mathcal{F}_d$ is detailed in the supplementary.

**Specular.** Our specular term is based on the Cook-Torrance model [6] which is generally defined as

$$f_s(\boldsymbol{\omega}_i, \boldsymbol{\omega}_o, r) = k_s \frac{D(\boldsymbol{\omega}_o, \boldsymbol{\omega}_i, r)\ G(\boldsymbol{\omega}_o, \boldsymbol{\omega}_i) F(\boldsymbol{\omega}_o, \boldsymbol{\omega}_i)}{4(\boldsymbol{n} \cdot \boldsymbol{\omega}_o)(\boldsymbol{n} \cdot \boldsymbol{\omega}_i)} \tag{4}$$

$$D(\cdot) = \alpha D_{12}(\cdot) + (1 - \alpha)D_{48}(\cdot) \tag{5}$$

where $k_s$ is the specular intensity, $D$ is the Normal Distribution Function (NDF), and $G$ is the masking and shadowing term, which is derived from the NDF [39]. $F$ models the Fresnel effect, for which we use Schlick's approximation [36]. As the NDF, we use the 2-Blinn-Phong-lobe mix introduced by Riviere et al. [33]. The advantage of this NDF representation is that roughness $r$ is a linear parameter, which is beneficial during the optimization.

Due to the ellipsoidal shape of 3D Gaussian primitives, it is non-trivial to associate a single normal vector to them. As suggested by Saito et al. [34], we observe that the normal varies with the viewing direction. We, therefore, use a second, smaller CNN $\mathcal{F}_v$, which takes expression features $\boldsymbol{f}_{expr}$ and the viewing direction $\boldsymbol{\omega}_o$, and predicts specular intensity $s_k$ and normal offsets $\delta\boldsymbol{n}$. While the general idea behind $\mathcal{F}_v$ is similar to [34], our U-Net architecture requires fewer network parameters, improving performance on consumer-level hardware. The final shading normals are obtained by adding the normal offsets $\delta\boldsymbol{n}_k$ to the mesh normals, followed by normalization. During training, we evaluate the specular term with the point light pattern of the current frame. For environment map relighting, we use the split-sum approximation [14]. We provide details in the supplementary.

### 4.4 Optimization

Given calibrated multi-view sequences from our dataset and corresponding estimated FLAME parameters, we jointly optimize $\mathcal{F}_g$, $\mathcal{F}_d$, $\mathcal{F}_v$, and static parameters $\boldsymbol{\mu}_k^{local}, \boldsymbol{a}_k, r_k$ with the following loss term:

$$\mathcal{L} = \mathcal{L}_{rgb} + \mathcal{L}_{reg} \tag{6}$$

$$\mathcal{L}_{reg} = \lambda_{normal}\mathcal{L}_{normal} + \lambda_{alpha}\mathcal{L}_{alpha} + \lambda_{scale}\mathcal{L}_{scale} + \lambda_{pos}\mathcal{L}_{pos} \tag{7}$$

where $\mathcal{L}_{rgb} = \lambda_{l1}\mathcal{L}_{l1} + \lambda_{SSIM}\mathcal{L}_{SSIM}$ is the photometric loss term consisting of an L1 and SSIM term as proposed by [15]. We set $\{\lambda_{l1}, \lambda_{SSIM}\}$ to $\{1.0, 0.2\}$ in all experiments. Our regularization loss $\mathcal{L}_{reg}$ consists of the normal loss $\mathcal{L}_{normal} = \|\delta\boldsymbol{n}\|$, which encourages the predicted normal offsets to be small, and thus, be close to the normals of the FLAME mesh. Our capture rig only contains lights on the frontal hemisphere, which would lead to artifacts when we render the avatars with lights from the rear or environment maps. We find that a simple L2 loss $\mathcal{L}_{alpha}$ between the rendered alpha maps and the alpha masks from background matting prevents the avatar from becoming too transparent. The scale loss is adapted from [34] and promotes the primitive scales to remain in a reasonable range. $\mathcal{L}_{pos}$ is another L2 term which drives $\mathcal{F}_g$ to predict small delta means. We set $\{\lambda_{alpha}, \lambda_{scale}, \lambda_{pos}\}$ to $\{2e{-}2, 2e{-}2, 1e{-}5\}$ in all experiments.

### 4.5 Implementation Details

We implement all networks and optimization logic in PyTorch [27], and write custom GPU kernels for the specular shading using the SLANG.D shading language [3]. For rendering the Gaussian primitives, we use *gsplat* [50]. We use a texture resolution of $512^2$ in all experiments, which results in 202k primitives after masking out texels from the FLAME UV map that are not assigned to any surface point. We use the 2023 version of FLAME [17] with the manually added teeth from Qian et al. [30]. We train our avatars at 1100x1604 resolution for 250k iterations with a batch size of 4, which takes approximately 30 hours on a single NVIDIA RTX A6000 GPU.

### 4.6 Differences to RGCA

RGCA [34] uses a variational autoencoder, which learns a personalized expression space and predicts the parameters of the precomputed radiance transfer function. In contrast, our method builds directly on top of FLAME [17], which has a shared expression space across identities, enabling applications

Table 2: Quantitative results on held-out lights on both the training and held-out segments.

| Method | Relighting | | | Relighting + Self-Reenactment | | |
|--------|-----------|-----------|-----------|-----------|-----------|-----------|
| | PSNR↑ | SSIM↑ | LPIPS↓ | PSNR↑ | SSIM↑ | LPIPS↓ |
| RGCA | 29.21 | 0.8462 | 0.1659 | 26.31 | 0.8206 | 0.1917 |
| RGCA$_{\text{FLAME}}$ | 29.78 | 0.8464 | 0.1444 | 26.91 | 0.8282 | 0.1667 |
| Ours | **31.38** | **0.8956** | **0.1040** | **28.08** | **0.8730** | **0.1317** |

such as cross-reenactment. Generally speaking, our avatars can be animated with FLAME parameters from any source without the need for a personalized encoder. Further, we model diffuse light transport with a small MLP and use a Cook-Torrance [6, 33] variant for specular reflection.

# 5 Experiments

We evaluate our method on 4 subjects from our dataset, where our focus lies on relighting and self-reenactment. From the 16 available camera views, we use 15 for training, and hold out the center camera for testing. We further hold out 4 light patterns from training altogether. From the available sequences, we use all scripted sequences for training and use the *free* sequence for testing. As the test metrics, we use the Peak-Signal-to-Noise Ratio (PSNR), Structural-Similarity-Index-Measure (SSIM) [42] and the Learned Perceptual Image Patch Similarity (LPIPS) [52]

**Baselines.** Our main baseline is *Relightable Gaussian Codec Avatars* (RGCA) [34], a recent method that builds head avatars by decoding learned expression codes to 3D Gaussian attributes for geometry and intrinsic radiance transfer. The resulting avatars can be relighted by integrating the predicted intrinsic radiance properties with novel light sources. The input to RGCA are the vertices of a coarse template mesh together with unwrapped average textures, for which we use the FLAME meshes and textures from the VHAP [29, 30] tracking. Since the expression space of RGCA is learned per identity, it requires comprehensive training sequences, while our method can leverage the existing FLAME expression space. Therefore, we introduce a second baseline denoted *RGCA$_{\text{FLAME}}$*, where we replace the learned expression latent space with FLAME expression coefficients.

## 5.1 Relighting and Self-Reenactment

Our primary target application is reenactment under novel illuminations. Therefore, we animate our trained avatars with the FLAME parameters of the held-out sequence and select those frames with a light pattern not seen during training. We then render the avatars from the held-out camera view. In Table 2 we report the quantitative results of relighting for both a training and test sequence. Figure 4 presents the qualitative results of our avatars rendered from the test camera, with an unseen lighting condition and expression. A comparison under environment map relighting is shown in Figure 7. Our rendered avatars match the target appearance more closely in terms of the color and fine geometric details, which enables more realistic specular reflections. Notably, we observe that RGCA conditioned on FLAME parameters performs strictly better than the original version with the personalized expression space. We hypothesize that learning an expression space per subject is suboptimal for reenactment tasks.

In Figure 5, we qualitatively compare the intrinsic decomposition performed by our approach to the baselines. Our method recovers cleaner albedo and sharper specular highlights from the data observations and faithfully decomposes the diffuse and specular parts of the material. In addition, we recommend to watch our supplementary video for more results, which allow for a more complete comparison, including the temporal axis.

## 5.2 Ablation Study

We verify the key components of our method with ablation experiments, which we conduct with the same subjects. A qualitative and quantitative comparison is presented in Figure 6 and Table 3, respectively.

**PBR.** We compare our full model to a version where we replace the hybrid neural shading with a classic PBR shading model using a Lambertian term for diffuse, and a Cook-Torrance [6] for specular reflection. This simple appearance model cannot reproduce the complex appearance of skin since subsurface scattering is not modeled, which results in synthetically looking renderings.

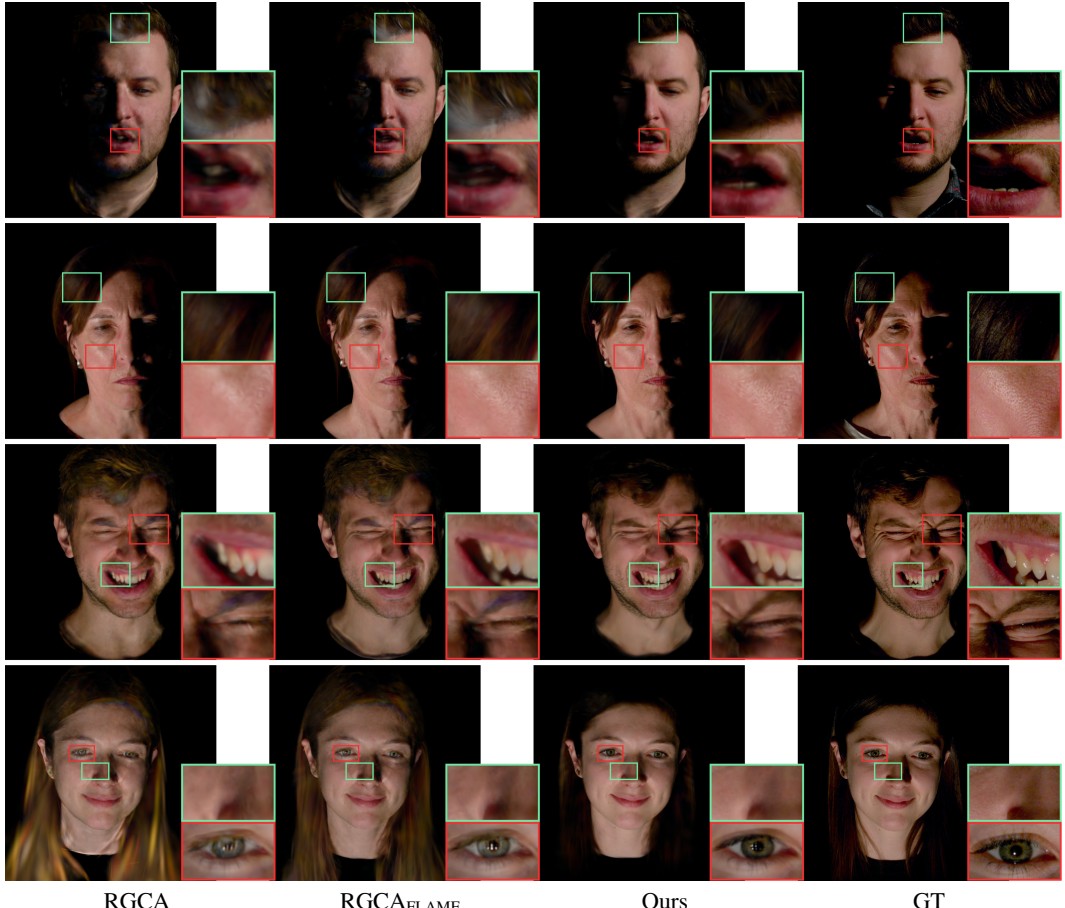

| RGCA | RGCA$_{\text{FLAME}}$ | Ours | GT |

Figure 4: **Relighting and Self-Reenactment**: Qualitative comparison on held-out segments and held-out illuminations.

**PRT Diffuse.** We compare our neural diffuse component to the learned precomputed radiance transfer (PRT) model introduced in [34]. The respective SH coefficients are directly predicts by $\mathcal{F}_g$. Although, we observe good results on the training frames, PRT struggles to generalize to novel illuminations, which aligns with our findings from Section 5.1.

**SG.** To justify the choice of the specular term, we ablate our specular BRDF to a simple Spherical Gaussian term. Compared to our Cook-Torrance term, we observe slightly better PSNR and LPIPS scores as well as marginal sharper details under point light illumination. However, as depicted in Fig. 6, we observe significantly more natural and detailed pore-level reflection with our Cook-Torrance based specular model.

**Alpha Loss.** The key component to prevent the avatars from becoming too transparent is the alpha loss using estimated foreground segmentation masks. Since our capture setup only has lights and cameras on the frontal hemisphere, we observe artifacts with environment map relighting when using no regularization. We want to highlight that this simple regularization scheme effectively reduces the complexity of our capture setup, the resulting dataset, and computational cost during training by half.

**Expression Features.** We compare our expression-dependent features against static feature vectors as proposed by Giebenhain et al. [10]. Here, we use static features for the diffuse BRDF network $\mathcal{F}_d$, and use the FLAME parameters as a condition for $\mathcal{F}_v$. We observe that without the expression-dependent features, the model fails to accurately reproduce pore-level details and specular highlights. We can further notice worse color and reflections compared to the full model.

## 5.3 Application

Once trained, the only inference parameters are FLAME expression and pose parameters, which can be obtained from monocular videos [29, 30]. We demonstrate this by animating our avatars with short

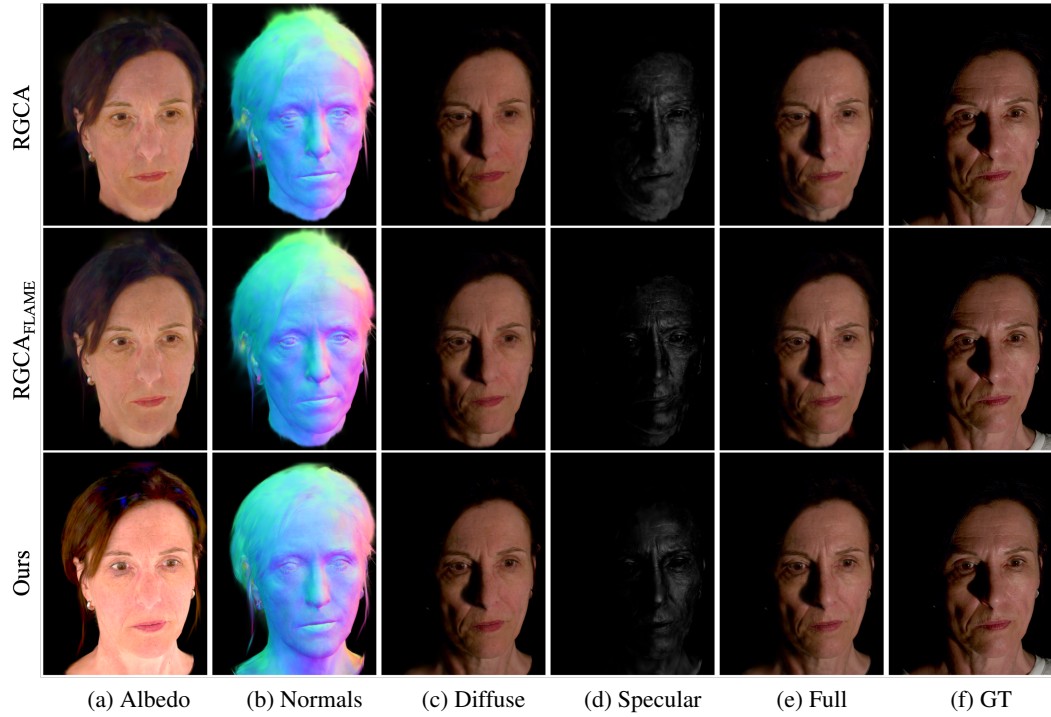

| | (a) Albedo | (b) Normals | (c) Diffuse | (d) Specular | (e) Full | (f) GT |

Figure 5: **Comparison of Intrinsic Decomposition**: We compare the recovered albedo (a) and normals (b), as well as the diffuse (c) and specular (d) contributions on a training frame that sum up to the final rendering (e). Note that the reference image (f) is identical in all rows.

Table 3: **Ablations**: We conduct ablations with the same 4 subjects and report relighting and reenactment results on the training and test expressions.

| Method | Relighting | | | Relighting + Self-Reenactment | | |
|---|---|---|---|---|---|---|
| | PSNR↑ | SSIM↑ | LPIPS↓ | PSNR↑ | SSIM↑ | LPIPS↓ |
| w/ PBR shading | 29.42 | 0.8719 | 0.1344 | 26.31 | 0.8448 | 0.1665 |
| w/ PRT diffuse | 29.23 | 0.8374 | 0.1577 | 25.47 | 0.8074 | 0.1918 |
| w/ SG | **31.55** | 0.8953 | **0.1031** | **28.09** | 0.8729 | **0.1310** |
| w/o alpha loss | 31.34 | 0.8955 | 0.1043 | 28.07 | 0.8729 | 0.1328 |
| w/o expr. features | 31.23 | 0.8928 | 0.1071 | 28.13 | 0.8717 | 0.1332 |
| Ours (full) | 31.38 | **0.8956** | 0.1040 | 28.08 | **0.8730** | 0.1317 |

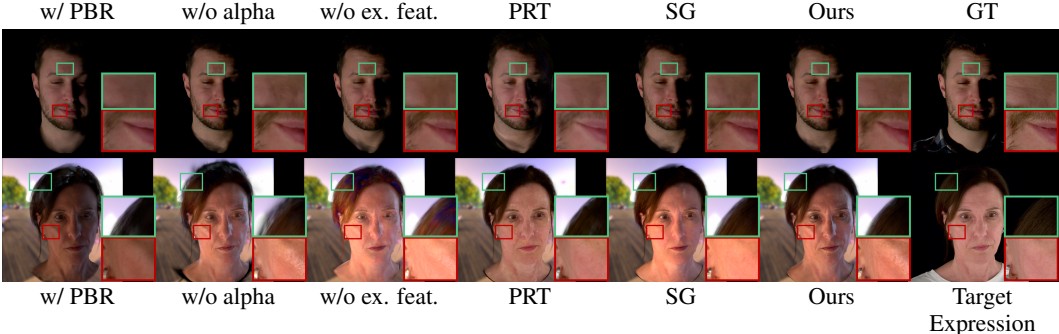

| w/ PBR | w/o alpha | w/o ex. feat. | PRT | SG | Ours | Target Expression |

Figure 6: **Ablation Study**: With only PBR shading, the avatar has a synthetic, plastic-like appearance. Without the alpha loss, we observe artifacts when rendering with environment maps. The expression-dependent features further improve both appearance and fine geometric details.

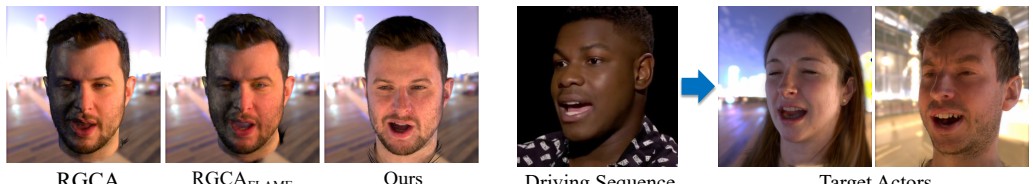

| RGCA | RGCA$_{\text{FLAME}}$ | Ours | Driving Sequence | Target Actors |

Figure 7: *Left:* Qualitative comparison on environment map relighting. *Right:* Animation using monocular videos.

Table 4: **Runtime comparison**: We report the component-wise inference time in milliseconds.

| Method | CNNs | Diffuse Shading | Specular Shading | Splatting | Total |
|--------|------|-----------------|------------------|-----------|-------|
| RGCA | 9ms | 1ms | 1ms | 9ms | 20ms |
| Ours | 4ms | 3ms | 1ms | 9ms | 17ms |

video sequences from the VFHQ dataset [44]. We obtain the FLAME parameter with the monocular version of the VHAP tracker [29], and relight our avatars with environment maps collected from PolyHaven [1]. We present the results in Figure 7 and highly encourage the reader to watch the accompanying video for temporal results.

### 5.4 Runtime

In Table 5, we summarize the runtime of the components of our method and compare it with RGCA [34]. We conducted all measurements on a single NVIDIA RTX A6000 GPU using a UV resolution of $512^2$, which results in 202k primitives (due to masking of texels not assigned to any surface location). We render images at a resolution of 1100x1604, corresponding to the training resolution of our avatars.

### 5.5 Discussion

**Limitations.** While our method delivers state-of-the-art results and enables more practical animation than previous methods, our approach is still not without limitations. While our capture setup is an order of magnitude more economical than existing setups [34, 4, 12], avatar training still requires several thousand frames and a diverse set of training expressions. Obtaining photorealistic avatars from causal phone captures with uncalibrated lighting remains an open challenge for future work. The FLAME base geometry is limited in its expressiveness, is sensitive to tracking failures (particularly with respect to gaze direction), and does not model the mouth interior. These limitations are consequently inherited by our avatars. As of now, the neural diffuse shading model is trained from scratch jointly with the avatar. Using our dataset to learn an appearance prior of human faces and heads is an interesting direction for future work.

**Ethical Considerations.** Creating photorealistic, relightable avatars entails the potential for various malicious use cases, such as identity theft, deepfakes, and privacy violations. This is a particular concern when avatars can be driven from simple video sequences, as in our case. However, to create an avatar with our method, the respective subject must first be scanned in our capture setup, which is only applied to a limited number of consenting individuals. Further, we will be restrictive with access to our dataset as outlined in Section 3.4.

## 6 Conclusion

We have presented *BecomingLit*, a novel framework for reconstructing photorealistic, relightable avatars from a capture setup, orders of magnitude more economical than previous state-of-the-art methods. We have proposed a new hybrid shading approach for 3D Gaussian primitives, which enables better generalization to novel illuminations and expressions. Our relightable avatars can be animated from simple videos and relighted with both point lights and environment maps. Along with our method, we will publish a new dataset of faces under OLAT conditions, which is unprecedented in terms of resolution and frame rate. We believe that this will democratize research on facial appearance modeling and serve as a valuable contribution to the community.

## Acknowledgments and Disclosure of Funding

This work was supported by the ERC Consolidator Grant Gen3D (101171131) and the German Research Foundation (DFG) Research Unit "Learning and Simulation in Visual Computing". Additionally, we would like to thank Angela Dai for the video voice-over.

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

# BecomingLit: Relightable Gaussian Avatars with Hybrid Neural Shading

# Supplementary Material

**Jonathan Schmidt**       **Simon Giebenhain**       **Matthias Nießner**
Technical University of Munich

`jonathsch.github.io/becominglit`

## A    Network Architecture

Our geometry module $\mathcal{F}_g$ maps FLAME [17] expression, jaw and eyes pose coefficients, and predicts per-texel attributes $\{\delta\boldsymbol{\mu}, \boldsymbol{q}, \boldsymbol{s}, \sigma, \boldsymbol{f}^{expr}\}_{k=1}^M$, where $\boldsymbol{f}_k^{expr}$ has a dimension of 32 in all experiments. We use the first 100 principal components for the expression parameter, and a rodrigues parameterization for jaw and both eye rotations. Hence, our input of shape $\mathbb{R}^{109}$ is transformed by a linear layer and then reshaped to $256 \times 8 \times 8$. A set of transposed convolutional layers then gradually upsamples the feature maps to the final output of shape $43 \times 512 \times 512$. We use leaky-ReLU as activation function for all layers except for the final output. For all convolutional layers, we adopt untied bias [4].

$\mathcal{F}_v$ takes as input the per-Gaussian feature map $\boldsymbol{f}^{expr}$, and the view direction, which is encoded using a single linear layer (8-dim output shape) and then expanded to the height and with dimension of the feature map. We concatenate the feature map and encoded view direction and feed it through a single convolutional layer which downsamples the input by half. Finally, a transposed convolutional layer maps the latent feature map back to its original resolution with 4 output channels.

Our diffuse BRDF network $\mathcal{F}_d$ is a 3-layer MLP with hidden dimension 64 and leaky-ReLU activation in every layer, except the last one. The input is the concatenation of $\boldsymbol{f}_k^{expr}$ and the spherical harmonics coefficients of the incident light.

## B    Environment Map Rendering

In this section, we provide further details on how we render our avatars with all-frequency continuous illumination in the form of environment maps. While our diffuse BRDF trivially adopts to continuous illumination due to the spherical harmonics parameterization, we need to adopt the specular shading of the primitives.

For the specular pre-integration, we follow the split-sum approximation [14]. Karis et al. [14] propose to assume that the view direction $\boldsymbol{\omega}_o$ and the surface normal $\boldsymbol{n}$ are identical. With that assumption, the specular reflection is no longer view-dependent, and we can pre-integrate the environment map for different roughness values using a mipmap. In each mipmap level, we numerically integrate $L_i$ with importance sampling using the Blinn-Phong distribution:

$$L_o^{specular}(\boldsymbol{x}, \boldsymbol{\omega}_o) = \int_\Omega L_i(\boldsymbol{\omega}_i) D(\boldsymbol{h}, \boldsymbol{n}, r^2)(\boldsymbol{\omega}_i \cdot \boldsymbol{n}) d\boldsymbol{\omega}_i * \int_\Omega k_s \frac{DGF}{4(\boldsymbol{\omega}_o \boldsymbol{n})(\boldsymbol{n}\boldsymbol{\omega}_i)} d\boldsymbol{\omega}_i \tag{8}$$

The incoming illumination $L_i(\boldsymbol{\omega}_i)$ is now stored in the pre-integrated environment map $\hat{L}_{specular}(\boldsymbol{\omega}, r)$. During rendering, we linearly interpolate the mip levels to obtain the final radiance value for the roughness parameter. Hence, the new specular term becomes:

$$L_o^{specular}(\boldsymbol{x}, \boldsymbol{\omega}_o) \approx \hat{L}_{specular}(\boldsymbol{\omega}, r) \int_\Omega k_s \frac{DGF}{4(\boldsymbol{\omega}_o \boldsymbol{n})(\boldsymbol{n}\boldsymbol{\omega}_i)} d\boldsymbol{\omega}_i \tag{9}$$

The remaining integral is essentially the integration of the BRDF with a completely white environment light. We can substitute the Fresnel term $F(\boldsymbol{\omega}_o, \boldsymbol{h})$ with the Schlick approximation [36] and factor out $F_0$:

| RGCA | RGCA (subset) | Ours | Ours (subset) | GT |

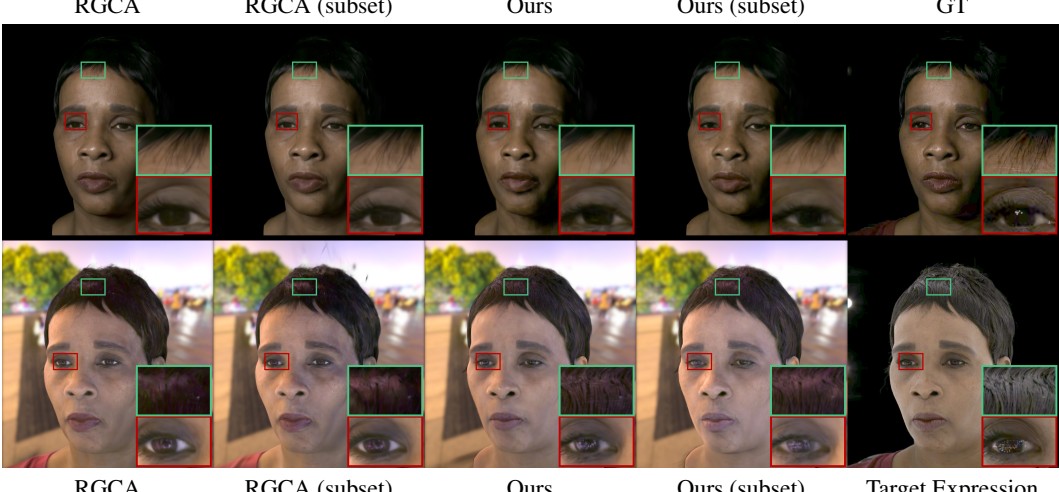

| RGCA | RGCA (subset) | Ours | Ours (subset) | Target Expression |

Figure 8: **Goliath-4 Evaluation**: Qualitative comparison on held-out frames and held-out light patterns from an unseen viewpoint.

$$\int_{\Omega} f_s(\boldsymbol{\omega}_i, \boldsymbol{\omega}_o)(\boldsymbol{n} \cdot \boldsymbol{\omega}_i)d\boldsymbol{\omega}_i = F_0 \int_{\Omega} \frac{f_s(\boldsymbol{\omega}_i, \boldsymbol{\omega}_o)}{F(\boldsymbol{\omega}_o, \boldsymbol{h}} \left(1 - (1 - \boldsymbol{\omega}_o \cdot \boldsymbol{h})^5\right)(\boldsymbol{\omega}_i \cdot \boldsymbol{n})d\boldsymbol{\omega}_i$$
$$+ \int_{\Omega} \frac{f_s(\boldsymbol{\omega}_i, \boldsymbol{\omega}_o)}{F(\boldsymbol{\omega}_o, \boldsymbol{h}}(1 - \boldsymbol{\omega}_o \cdot \boldsymbol{h}^5(\boldsymbol{\omega}_i \cdot \boldsymbol{n})d\boldsymbol{\omega}_i \quad (10)$$

These integrals depend on the two inputs $(\boldsymbol{\omega}_i \cdot \boldsymbol{n})$ and the roughness parameter $r$ and act as a scale and bias to $F_0$. We pre-integrate both terms for all possible input combinations in $[0, 1]^2$ and store the two outputs in the 2D texture map $\hat{\boldsymbol{f}}_s(\boldsymbol{\omega}, r)$

During rendering, we can now compute the shaded color by evaluating the following terms:

$$\boldsymbol{\omega}_r = -\boldsymbol{\omega}_o - 2(-\boldsymbol{\omega}_o \cdot \boldsymbol{n})\boldsymbol{n} \quad (11)$$

$$a, b = \hat{\boldsymbol{f}}_s((\boldsymbol{\omega}_r \cdot \boldsymbol{n}), r) \quad (12)$$

$$c_k^s(\boldsymbol{\omega}_o) = (a\ k_s + b)\hat{L}_{specular}(\boldsymbol{\omega}_r, r) \quad (13)$$

## C   Goliath-4 Dataset

To demonstrate that our method also generalize well to data domains beyond our new light stage dataset, we perform additional experiments on the Goliath-4 dataset [22]. We train both our method and RGCA [34] in two different configurations. (1) With the full available set of cameras, holding out 10 random views for evaluation, and (2) with a random subset of 16 train cameras and 4 cameras for evaluation. We limit ourselves to one of the subjects (QZX685) and use the provided train/test split for evaluating unseen expressions and hold out a random subset of 10% of the available light patterns to evaluate relighting capabilities. Our method consistently outperforms RGCA on the SSIM and LPIPS metrics. We report quantitative results in Table 5, and a qualitative comparison in Figure 8. We want to point out, that the publicly released dataset [22] is subsampled to every 10th frame, and the provided images are heavily compressed, which inevitably leads to a drop in quality compared to the results shown in [34].

## D   Capture Script

For each participant of our dataset, we record 7 sequences in total. The first 6 consist of a predefined set of facial expressions, emotions and sentences that we ask the subjects to perform and read out. In the 7th sequence the participant is free to perform any facial expression for 20s. The instructions are given via a screen that is placed

Table 5: **Goliath-4 Dataset:** Quantitative results on Self-Reenactment and Relighting. We evaluate self-reenactment on the validation camera views.

| Method | Full Cam Set | | | Random Cam Subset (10%) | | |
|--------|-------|-------|--------|-------|-------|--------|
| | PSNR↑ | SSIM↑ | LPIPS↓ | PSNR↑ | SSIM↑ | LPIPS↓ |
| RGCA | **29.89** | 0.8869 | 0.1392 | **28.74** | 0.8753 | 0.1468 |
| Ours | 29.70 | **0.9080** | **0.1165** | 28.68 | **0.8966** | **0.1298** |

in front of the subject. In the following we provide a list of the single components, which during the capture sessions are accompanied with images.

- **Expressions-1**:
    - Head rotation with mouth open and closed
    - Eyes blink
    - Eyes squint
    - Eyebrows up / down
    - Puffed Cheeks
    - Mouth Vacuum
    - Nose Wrinkle
    - Lip bite
- **Expressions-2**
    - Grin (multiple variations)
    - Jaw movement
    - Lip licking
    - Tongue
- **Emotions**
    - Shout
    - Laugh
    - Surprise
    - Fear
    - Angry
    - Sad
    - Disgust
    - Happy
    - Confusion
    - Amazement
    - Embarrassment
- **Sentences-1**
    - A cramp is no small danger on a swim.
    - He said the same phrase thirty times.
    - Pluck the bright rose without leaves.
    - Two plus seven is less than ten.
    - The glow deepened in the eyes of the sweet girl.
    - By eating yogurt you may live longer.
- **Sentences-2**
    - Bring your problems to the wise chief.
    - Write a fond note to the friend you cherish.
    - Clothes and lodging are free to new men.
    - We frown when events take a bad turn.
    - Port is a strong wine with a smoky taste.
    - They had slapped their thighs.
- **Sentences-3**

- She always jokes about too much garlic in his food.
- Why put such a high value on being top dog.
- All your wishful thinking won't change that.
- Take charge of choosing her bridesmaids gowns.
- Why buy oil when you always use mine.

