# OpenReview forum: "BecomingLit: Relightable Gaussian Avatars with Hybrid Neural Shading"
_NeurIPS.cc/2025/Conference — NeurIPS 2025 poster_

### Official Review · Reviewer_B4Yt · 2025-06-21

**Clarity:** 3
**Significance:** 2
**Originality:** 2
**Rating:** 4
**Confidence:** 4

**Summary:**

The paper introduces a novel dataset and a method that utilizes it, achieving state-of-the-art results. The introduced dataset is captured using a lightweight capture system consisting of 16 cameras and 40 lights, which is significantly smaller compared to, for instance, the RGCA dataset. Despite its smaller size, the quality provided by the neural appearance representation is very high, producing sharp and detailed avatars.

**Questions:**

*RGCA* and *BecomingLit* use different image formation models. From the initial evaluation, it appears that using the classical Cook-Torrance microfacet model for specular reflection gives superior results. However, I would like to see whether this holds true when both methods are trained on the same dataset. This is an interesting aspect that I believe could be better investigated. Specifically, the comparison between RGCA's linear radiance transfer and the small neural BRDF function used in BecomingLit to model light transport, without the need to store it per Gaussian primitive, deserves further analysis.

I would suggest performing two additional evaluations: one on the full RGCA dataset using all 144 cameras, and another on a subsampled set of cameras. The drop in quality for RGCA likely also depends on the tracking. However, for the Goliath experiments, I would recommend using the original setup (with the provided tracking) and only modifying the number of views used. There may still be some inconsistencies, since I assume BecomingLit would need to use VHAP tracking on Goliath to remain compatible, but this would still provide more insight into the underlying model expressiveness.

**Questions**
1) Do you use any regularizers on the prediction of $F_v$ for Cook–Torrance formula?
2) What is the motivation behind using the classical Cook–Torrance microfacet model for specular?

**Ethical Concerns:**

["NO or VERY MINOR ethics concerns only"]

**Final Justification:**

After reading the rebuttal, most of my concerns were clarified, and a new ablation was provided. I will maintain my previous score of Weak Accept. However, I would like to ask authors to attach the new findings in the final version of the paper.

**Limitations:**

The quality is still lower compared to RGCA trained on the Goliath dataset, especially in terms of teeth and hair quality. However, it remains unclear how much of this issue is due to the dataset or the image formation model used.

**Quality:**

3

**Strengths And Weaknesses:**

**Weaknesses**

1) The provided evaluation is not sufficient. While I can imagine that RGCA does not perform well under few-view settings, I still have doubts about the presented quality; it appears significantly worse than in the original paper. I would suggest running BecomingLit on the RGCA-released dataset as well, to enable a fair comparison.
2) There are still artifacts, especially when relighting dynamic facial parts such as the teeth.

**Strenghts**

1) Very high-quality results are obtained on a dataset that is orders of magnitude smaller compared to other state-of-the-art methods. The dataset will be publicly available, which is a valuable contribution to the community and will support further research on relightable avatars.
2) The presented appearance model is an interesting alternative to the fully linear RGCA approach. RGCA uses a fully linear model for both specular and diffuse lighting, utilizing spherical harmonics for diffuse and spherical Gaussians for specular components, which allows for explicit light transfer. In the context of this work, the diffuse color (BRDF) is represented by a small neural network conditioned on lighting and expressions. The specular component, on the other hand, uses the Cook-Torrance microfacet model, where $k_s$ is predicted by a neural network $F_v$, along with normal perturbations $\delta n_k$. The roughness $r_k$ is learned per Gaussian as a variable. This is an elegant formulation based on a widely used specular model from computer graphics. However, the effectiveness of this formulation remains to be seen, as I believe the presented results are not sufficient to determine which aspects contribute most to the performance.

---

> ### Author Rebuttal · Authors · 2025-07-31
>
> >The provided evaluation is not sufficient. While I can imagine that RGCA does not perform well under few-view settings, I still have doubts about the presented quality; it appears significantly worse than in the original paper.
>
> RGCA results from the original paper were obtained using a capture setup with 160 cameras and 460 lights densely distributed on a 360-degree dome structure. Our comparisons were conducted with data from our capture setup, consisting of (only) 16 cameras and 40 lights, and covering only the frontal hemisphere around the head.
>
> Another difference is the evaluation protocol. While we test and visualize novel-view synthesis, unseen illuminations, and reenactment jointly in Table 2 and Figure 4, RGCA evaluates each modality separately, e.g., Figure 9 of [34] shows novel-view synthesis on a training frame and Figure 10 in [34] shows relighting on a training view and training expression which is a much less challenging setting than in our evaluation. Also, while we hold out a set of lights from training altogether, RGCA only excludes a set of light patterns, which, due to the additive nature of light, is a less meaningful test. Still, we indeed acknowledge that RGCA can produce higher-fidelity results when trained with more camera views and light patterns.
>
> >I would suggest performing two additional evaluations: one on the full RGCA dataset using all 144 cameras, and another on a subsampled set of cameras. The drop in quality for RGCA likely also depends on the tracking. However, for the Goliath experiments, I would recommend using the original setup (with the provided tracking) and only modifying the number of views used. There may still be some inconsistencies, since I assume BecomingLit would need to use VHAP tracking on Goliath to remain compatible, but this would still provide more insight into the underlying model expressiveness.
>
> We present the results of the additional evaluation on the Goliath-4 dataset below. Note, however, that the publicly released dataset is subsampled to every 10th frame, and the provided images are heavily compressed. The required FLAME tracking we computed for the Goliath-4 dataset is also not perfectly accurate due to the limited time of the rebuttal period.  We train both our method and RGCA in three different configurations. (1) With the full available set of cameras, holding out 10 random views for evaluation, (2) with a random subset of 16 train cameras and 4 cameras for evaluation, and (3), with a subset of 16 train and 4 test cameras sampled from the frontal region to mimic a similar setting than in our capture setup.
>
> Due to the limited time, we limit ourselves to one of the subjects (QZX685) and use the provided train/test split for evaluating unseen expressions and hold out a random subset of 10% of the total light patterns to evaluate relighting capabilities. Our method consistently outperforms RGCA on the SSIM and LPIPS metrics. We will add a qualitative comparison to the final version of the paper.
>
> #### Full Set of Cameras
>
> | Method             | PSNR  | SSIM   | LPIPS  |
> |--------------------|:-----:|:------:|:------:|
> | RGCA               | 29.89 | 0.8869 | 0.1392 |
> | Ours               | 29.70 | 0.9080 | 0.1165 |
>
> #### Random Camera Subset (10%)
>
> | Method             | PSNR  | SSIM   | LPIPS  |
> |--------------------|:-----:|:------:|:------:|
> | RGCA               | 28.74 | 0.8753 | 0.1468 |
> | Ours               | 28.68 | 0.8966 | 0.1298 |
>
> #### Frontal Camera Subset (10%)
>
> | Method             | PSNR  | SSIM   | LPIPS  |
> |--------------------|:-----:|:------:|:------:|
> | RGCA               | 29.88 | 0.8922 | 0.1326 |
> | Ours               | 29.82 | 0.9137 | 0.1139 |
>
> >RGCA and BecomingLit use different image formation models. From the initial evaluation, it appears that using the classical Cook-Torrance microfacet model for specular reflection gives superior results. However, I would like to see whether this holds true when both methods are trained on the same dataset. This is an interesting aspect that I believe could be better investigated. Specifically, the comparison between RGCA's linear radiance transfer and the small neural BRDF function used in BecomingLit to model light transport, without the need to store it per Gaussian primitive, deserves further analysis.
>
> First, we want to clarify that both methods were trained on the same dataset for all comparisons we made in the paper.
>
> In addition to the evaluations on the Goliath-4 dataset shown above, we also present the results of two additional ablations to further justify the design of our appearance model. The results are extensions to the ablation study in Table 3 of the main paper and follow the same evaluation protocol. For completeness, we also evaluate novel-view synthesis on the training frames.
>
> #### NVS
>
> | Method             | PSNR  | SSIM   | LPIPS  |
> |--------------------|:-----:|:------:|:------:|
> | Ours (Full)        | 32.12 | 0.9266 | 0.0699 |
> | Spherical Gaussians | 31.86 | 0.9254 | 0.0700 |
> | PRT Diffuse        | 30.70 | 0.9031 | 0.0987 |
>
> #### NVS + Relighting
>
> | Method             | PSNR  | SSIM   | LPIPS  |
> |--------------------|:-----:|:------:|:------:|
> | Ours (Full)        | 31.38 | 0.8956 | 0.1040 |
> | Spherical Gaussians | 31.55 | 0.8953 | 0.1031 |
> | PRT Diffuse        | 29.23 | 0.8374 | 0.1577 |
>
> #### NVS + Relighting + Self-Reenactment
>
> | Method             | PSNR  | SSIM   | LPIPS  |
> |--------------------|:-----:|:------:|:------:|
> | Ours (Full)        | 31.38 | 0.8956 | 0.1040 |
> | Spherical Gaussians | 31.55 | 0.8953 | 0.1031 |
> | PRT Diffuse        | 29.23 | 0.8374 | 0.1577 |
>
> **Spherical Gaussians Specular**: In this experiment, we replace our Cook-Torrance specular term with the spherical Gaussians formulation from RGCA [34], leaving the rest of our method untouched. While we observe slightly better results in almost all metrics, we still decided in favor of the Cook-Torrance model since we observed slightly sharper specular highlights under environment relighting.
>
> **Precomputed Radiance Transfer (PRT)**: Here, we replace our neural diffuse BRDF with the PRT model from RGCA [34]. Therefore, we modify our dynamics model F_g to also predict the spherical harmonics coefficients for the PRT function of the primitives, which we then integrate with the incident lighting. While we observe reasonable results under training illumination, PRT struggles to generalize to novel illuminations, which aligns with our observations from the comparisons with RGCA in Table 2 and Figures 4 and 7.
>
> We will extend Table 3 with the results from above and also add qualitative comparisons of both ablations in Figure 6 in the final version of the paper.
>
> >There are still artifacts, especially when relighting dynamic facial parts such as the teeth.
>
> Reconstructing the mouth interior is particularly challenging, due to its highly non-rigid deforming geometry. Further, the mouth interior is only observed under a limited number of expressions and camera views due to occlusions. We tackled these challenges by adding explicit teeth geometry to the FLAME template mesh, which spawns additional Gaussian primitives for this. As stated in the main paper, we suspect that the described artifacts mainly arise from the limited expressiveness of the FLAME model. We believe that our method would deliver even better results when built with a more expressive parametric model. We will state this limitation more clearly in the final version of the paper.
>
> >Do you use any regularizers on the prediction of  for Cook–Torrance formula?
>
> In the initial phase of the optimization, we regularize the predicted normal offsets to be small, thus the primitive’s normal remains close to the mesh normal. We gradually relax this constraint during the first 100k iterations of training. The roughness and visibility parameters are not regularized explicitly, but we use a sigmoid activation to ensure their values stay within the valid range.
>
> >What is the motivation behind using the classical Cook–Torrance microfacet model for specular?
>
> Using the Cook-Torrance model, we can leverage the specular normal distribution function (NDF) from [33], which gives superior results in the environment map relighting setting compared to a simple Gaussian distribution. Please also refer to our additional ablation on the appearance model from above.

---

> > ### Author Response · Authors · 2025-08-04
> > **Correction of Our Rebuttal Ablation Results**
> >
> > Dear Reviewer,
> >
> > we noticed that we accidentally posted tables with duplicate entries for our additional ablation results. We sincerely apologise for this mistake and want to correct it here:
> >
> > #### NVS
> >
> > | Method             | PSNR  | SSIM   | LPIPS  |
> > |--------------------|:-----:|:------:|:------:|
> > | Ours (Full)        | 32.12 | 0.9266 | 0.0699 |
> > | Spherical Gaussians | 31.86 | 0.9254 | 0.0700 |
> > | PRT Diffuse        | 30.70 | 0.9031 | 0.0987 |
> >
> > #### NVS + Relighting
> >
> > | Method             | PSNR  | SSIM   | LPIPS  |
> > |--------------------|:-----:|:------:|:------:|
> > | Ours (Full)        | 31.38 | 0.8956 | 0.1040 |
> > | Spherical Gaussians | 31.55 | 0.8953 | 0.1031 |
> > | PRT Diffuse        | 29.23 | 0.8374 | 0.1577 |
> >
> > #### NVS + Relighting + Self-Reenactment
> >
> > | Method             | PSNR  | SSIM   | LPIPS  |
> > |--------------------|:-----:|:------:|:------:|
> > | Ours (Full)        | 28.08 | 0.8730 | 0.1317 |
> > | Spherical Gaussians | 28.09 | 0.8729 | 0.1310 |
> > | PRT Diffuse        | 25.47 | 0.8074 | 0.1918 |

---

> > ### Author Response · Authors · 2025-08-05
> >
> > Dear Reviewer B4Yt,
> >
> > Thank you for your constructive comments and insightful suggestions, which have helped refine our work. The following summarizes your concerns, which we hope our rebuttal addresses appropriately.
> >
> > **RGCA evaluation and performance**: Besides our smaller capture setup compared to RGCA, we also use different evaluation protocols, jointly assessing novel-view synthesis, unseen illumination, and reenactment — a more challenging setting than RGCA’s. We performed additional experiments on the Goliath-4 dataset with multiple camera configurations, where our method consistently outperforms RGCA in SSIM and LPIPS metrics.
> >
> > **Comparison of reflectance models**: We conducted ablation studies comparing our neural BRDF and Cook-Torrance model against RGCA’s linear radiance transfer (PRT) and spherical Gaussians. The PRT-based model performed reasonably well under training conditions but struggled to generalize to novel lighting.
> >
> > **Relighting dynamic parts**: We tackled the challenges of reconstructing the mouth's interior by incorporating explicit tooth geometry, stating that a more expressive parametric model could further minimize artifacts.
> >
> > **Regularization of Cook–Torrance**: We apply soft regularization on normal offsets during early training and bound roughness and visibility using sigmoid activations.
> >
> > **Motivation for Cook–Torrance**: We chose Cook-Torrance for its superior performance in environment map relighting, particularly due to its more expressive specular NDF.
> >
> > We would be happy to provide further clarification or address any additional questions.

---

### Official Review · Reviewer_CBwD · 2025-07-01

**Clarity:** 3
**Significance:** 3
**Originality:** 2
**Rating:** 4
**Confidence:** 5

**Summary:**

The paper  proposes a method for reconstructing photo-realistic and relightable avatars from a ‘lightweight’ capture setup equipped with OLAT. First FLAME fitting is obtained from fully lit images of the lightstage. Then an expression dependent network is trained to predict gaussian attributes and an additional features vector. This feature vector is used to learn an implicit diffuse radiance with an explicit analytical specular BRDF.
Compared to Codec Avatars from Meta (RGCA) on the proposed dataset, the method outperforms RGCA especially in relighting.

**Questions:**

My major concerns are the following:

1. It is on the scientific contribution and the positioning of the paper with respect to RGCA. It is clear that this work is built on top of RGCA, however how this differs (other than using the implicit diffuse component) needs to be stated clearly. And it seems that this implicit diffuse term is the only difference which makes the scientific contribution limited to me. Also, reading the related work does not help positioning the paper and this needs to be improved. Also the claims provided in the introduction needs to be more specific to highlight the paper contribution

2. Using an hybrid reflectance field (with diffuse term learned implicitly) may violate energy conservation and the method does not seem to handle this which may lead to some saturation or artefacts. This need to be discussed as well

3. The RGCA works show their method still performs well with a low number of cameras. However results shown here for RGCA (in relighting) look completely off.  Especially that fig 8 in supp shows fair decomposition from RGCA. This needs to be discussed as well.

4. The predicted normals are tight to FLAME normals by design using a soft constraint. Is this enough to capture fine grained details such as wrinkles/folds in the normal map ?showing predicted normals map can be very insightful.

I like the results shown in the paper, also releasing a dataset with OLAT if of great value for the community.  However I am more concerned about the contributions itself. I would like to hear from authors and see other reviewers before making the final decision

**Ethical Concerns:**

["NO or VERY MINOR ethics concerns only"]

**Final Justification:**

I would like to thank the authors for addressing my concerns in the rebuttal. While I agree that the scientific contribution is a bit incremental with respect to RCGA. However, given the extended experimental protocol provided in the rebuttal, good paper flow, nice results and the release of the OLAT dataset (which is very valuable for the community), I vote for accepting this paper.

**Limitations:**

yes

**Quality:**

3

**Strengths And Weaknesses:**

The paper reads very well and the problem addressed here is well sound and very active. The release of OLAT dataset is highly appreciated by the community. The idea of using a hybrid BRDF material with only diffuse  components learned implicitly seems effective and allows the network to complete the cook-torrance model by learning global illumination effects.

---

> ### Author Rebuttal · Authors · 2025-07-31
>
> >It is on the scientific contribution and the positioning of the paper with respect to RGCA. It is clear that this work is built on top of RGCA, however how this differs (other than using the implicit diffuse component) needs to be stated clearly. And it seems that this implicit diffuse term is the only difference which makes the scientific contribution limited to me.Also, reading the related work does not help positioning the paper and this needs to be improved. Also the claims provided in the introduction needs to be more specific to highlight the paper contribution
>
> Our high-level goal is to reconstruct photorealistic, relightable head avatars that can be animated with novel expressions in real-time. In contrast to previous work, we focus on achieving this with a much more affordable capture setup, which we use to collect a dataset for the research community. Below, we detail the differences between our method and RGCA.
>
> RGCA is based on a variational Autoencoder (VAE), which learns a **personalized expression space** from dense multi-view captures of a light stage setting. The input to the encoder are the vertices of a **non-rigidly registered** template mesh, with corresponding texture maps. The decoders predict the geometric properties of Gaussian primitives along with the parameters for the precomputed radiance transfer function and specular shading.
>
> In contrast, our method builds directly on top of the **parametric** morphable model FLAME, which has a **shared expression space** across different identities. This key difference enables applications such as cross-reenactment, where expressions can be transferred between subjects, or generally speaking, an avatar can be animated with FLAME parameters from any source without the need for a personalized encoder. Therefore, we show animation from monocular videos, which is not possible with RGCA’s VAE approach.
>
> Given these FLAME parameters, we predict expression-dependent offsets for Gaussian primitives rigged to the base mesh, and a set of expression features for our appearance model, which computes diffuse reflectance with a small MLP and specular reflectance using view-dependent normals and visibility in a Cook-Torrance model. As stated in the main paper, we acknowledge that obtaining the parameters for the specular shading is inspired by RGCA [34].
>
> We will clarify the differences to RGCA as stated above in the final version of the paper, and also update the introduction to state our contributions more comprehensively.
>
> >Using an hybrid reflectance field (with diffuse term learned implicitly) may violate energy conservation and the method does not seem to handle this which may lead to some saturation or artefacts. This need to be discussed as well
>
> We decided to use a learned reflectance model in favor of a physically-based model since human skin consists of several different layers, each of which must be described with a subsurface scattering model. This would not only be hard to optimize but also impossible to render in real time. Further, our appearance model must not only cope with skin, but also with hair, eyeballs, the mouth interior, and teeth. Our ablation study in Table 3 and Figure 6 shows that a simple PBR shading model, as used in real-time graphics today, is not sufficient for representing these materials realistically.
>
> Our neural diffuse BRDF model does indeed not guarantee energy conservation. However, since it is trained on several thousand image observations, we conclude that fundamental characteristics of reflectance models, such as energy conservation, are learned implicitly to an acceptable level. We will point this out in the final version of the paper.
>
> Learning an appearance prior of human heads is an exciting direction for future work, and we expect that the release of our dataset opens up research in this direction for a broader community.
>
> >The RGCA works show their method still performs well with a low number of cameras. However results shown here for RGCA (in relighting) look completely off. Especially that fig 8 in supp shows fair decomposition from RGCA. This needs to be discussed as well.
>
> The RGCA results from the original paper were obtained using a capture setup with 160 cameras and 460 lights densely distributed on a 360-degree dome structure. As you mentioned, their ablation shows that it also works reasonably well with a small random subset of cameras (Fig 9 of [34]). However, our comparisons were conducted with data from our capture setup, consisting of 16 cameras and 40 lights, covering **only the frontal hemisphere** around the head.
>
> Another difference is the evaluation protocol. While we test and visualize novel-view synthesis, unseen illuminations, and reenactment jointly in Table 2 and Figure 4, RGCA evaluates each modality separately, e.g., Figure 9 of [34] shows novel-view synthesis on a training frame and Figure 10 in [34] shows relighting on a training view and training expression which is a much less challenging setting than in our evaluation. Also, while we hold out a set of lights from training altogether, RGCA only excludes a set of light patterns, which, due to the additive nature of light, is a less meaningful test. Still, we indeed acknowledge that RGCA can produce higher-fidelity results when trained with more camera views and light patterns.
>
> Figure 8 shows that RGCA fairly recovers albedo and normals. However, in our experiments, we found that the shown artifacts primarily arise from the learned diffuse PRT function, which tends to overfit to the training expressions and illuminations.
>
> Below, we present the results of an additional evaluation on the Goliath-4 dataset, which is closer to the original training setting of RGCA. Note, however, that the publicly released dataset is subsampled to every 10th frame, and the provided images are heavily compressed. The required FLAME tracking we computed for the Goliath-4 dataset is also not perfectly accurate due to the limited time of the rebuttal period. We train both our method and RGCA in three different configurations. (1) With the full available set of cameras, holding out 10 random views for evaluation, (2) with a random subset of 16 train cameras and 4 cameras for evaluation, and (3), with a subset of 16 train and 4 test cameras sampled from the frontal region to mimic a similar setting than in our capture setup.
>
> Due to the limited time, we limit ourselves to one of the subjects (QZX685) and use the provided train/test split for evaluating unseen expressions and hold out a random subset of 10% of the total light patterns to evaluate relighting capabilities. Our method consistently outperforms RGCA on the SSIM and LPIPS metrics. We will add a qualitative comparison to the final version of the paper.
>
> #### Full Set of Cameras
>
> | Method             | PSNR  | SSIM   | LPIPS  |
> |--------------------|:-----:|:------:|:------:|
> | RGCA               | 29.89 | 0.8869 | 0.1392 |
> | Ours               | 29.70 | 0.9080 | 0.1165 |
>
> #### Random Camera Subset (10%)
>
> | Method             | PSNR  | SSIM   | LPIPS  |
> |--------------------|:-----:|:------:|:------:|
> | RGCA               | 28.74 | 0.8753 | 0.1468 |
> | Ours               | 28.68 | 0.8966 | 0.1298 |
>
> #### Frontal Camera Subset (10%)
>
> | Method             | PSNR  | SSIM   | LPIPS  |
> |--------------------|:-----:|:------:|:------:|
> | RGCA               | 29.88 | 0.8922 | 0.1326 |
> | Ours               | 29.82 | 0.9137 | 0.1139 |
>
> >The predicted normals are tight to FLAME normals by design using a soft constraint. Is this enough to capture fine grained details such as wrinkles/folds in the normal map ?showing predicted normals map can be very insightful.
>
> Similar to normal maps in mesh rendering, our method uses the coarse FLAME mesh normals as base normals, and predicts expression and view-dependent offsets for the individual primitives. In addition, the $F_g$ CNN also predicts expression-dependent offsets for the primitive positions. Both combined can faithfully model wrinkles and folds of the skin beyond the capabilities of an ordinary normal map. Examples of the rendered normals can be found in Figure 5b. We see that the example shown is not ideal for highlighting the method’s capabilities with respect to modeling the respective effects, and we will include a more vivid example in the final version of the paper.

---

> > ### Author Response · Authors · 2025-08-05
> >
> > Dear Reviewer CBwD,
> >
> > We appreciate the time and effort you dedicated to reviewing our work. Your feedback provides valuable insights that have helped us improve our submission. We have outlined your concerns below and hope our rebuttal adequately addresses them.
> >
> > **Scientific contribution and differences from RGCA**: We clarified key differences from RGCA, including using the FLAME model with a shared expression space, which enables cross-subject reenactment — impossible with RGCA’s personalized VAE approach.
> >
> > **Reflectance model and energy conservation**: We opted for a learned reflectance model to handle the complexity of skin, hair, eyes, and teeth while enabling real-time rendering. While our neural diffuse BRDF doesn’t enforce energy conservation explicitly, training on large-scale image data implicitly captures physically plausible behavior.
> >
> > **RGCA performance with limited cameras**: We emphasize that we conduct our comparisons using a significantly lower-budget capture setup. Further, evaluation protocols differ — our experiments jointly test novel expressions, views, and lighting, while RGCA tests each modality separately on seen data. Our new evaluation shows that our method outperforms RGCA on the Goliath-4 dataset on SSIM and LPIPS metrics.
> >
> > **Normal prediction and fine details**: Our method builds on FLAME normals with predicted expression- and view-dependent offsets that enable modeling fine-grained skin details like wrinkles.
> >
> > Please let us know if anything needs further clarification or if there are any additional points you’d like us to address.

---

### Official Review · Reviewer_rxBB · 2025-07-01

**Clarity:** 4
**Significance:** 3
**Originality:** 2
**Rating:** 4
**Confidence:** 4

**Summary:**

Authors build a pipeline for relightable avatars acquired with a low cost light stage setup. They collect subjects under multiple viewpoints, illuminations and different facial expressions. Their model is based on a head parametric model (FLAME), which is further optimized to capture the real subject geometry. Their shading model is hybrid, using a neural diffuse BRDF combined with an analytical specular term. They prove high quality results under different conditions. Their dataset is publicly available to accelerate the research in that direction.

**Questions:**

My main question for authors would be to better clarify the differences with [34], since their explanation is simply “we prefer a hybrid architecture for the rendering”, and also why their comparisons with [34] do not seem to match what authors showed in the original paper.

Finally I’d like them to discuss more around the limitations I pointed out in the next section.

**Ethical Concerns:**

["NO or VERY MINOR ethics concerns only"]

**Final Justification:**

I thank the authors for the additional experiments and clarification, I am ok raising my rating to Borderline Accept. I would recommend add in the Limitations sections the ones I have suggested, in particular the challenges with real-world captures where users do not have access to an OLAT setup when building an avatar.

**Limitations:**

Authors discuss some limitations but they did not mention the following:

Density of the lighting setup: while the setup is indeed less expensive than a full light stage, the light density seems to cause issues, in particular OLAT like reflections can be observed when the subjects are relit (see video) or when predicting albedo (Fig 5). Authors should discuss this limitation and how they plan to address it.

Albedo prediction is a simpler problem when starting with OLAT images, but in practice, this is not the actual problem to solve. Indeed a more challenging case that authors did not address is how to build an avatar from any arbitrary lighting condition, which would mimic real-world scenarios. Also note that the albedo results show issues specifically around the eyes, where OLAT reflections can be observed (Figure 5)

**Quality:**

3

**Strengths And Weaknesses:**

[Strength] Relevant problem, well explained by authors and how they solve it
[Strength] Dataset release to accelerate the research in relightable avatars
[Strength] The method is simple yet effective and authors do a good job at explaining all the details.
[Strength] Well presented video results

[Weakness] My main criticism is around novelty and specifically comparing it with [34], while I understand the difference in terms of architecture (explicit pre-computed radiance transfer function vs hybrid), I did not find enough evidence by authors why this is needed.  Additionally, the comparisons authors show here do not seem to align with the results presented in [34] (see my next point)

[Weakness] Video results show limitations of the method, in particular around eyes. HDRI results show “OLATs” like reflections. The comparisons with Relightable Gaussian Codec Avatars [34] do not seem to match the original results in the paper. Indeed RGCA demonstrates high quality HDRI results, whereas authors mention in their video that the previous method “struggles” in that scenario.

---

> ### Author Rebuttal · Authors · 2025-07-31
>
> >My main question for authors would be to better clarify the differences with [34], since their explanation is simply “we prefer a hybrid architecture for the rendering”, and also why their comparisons with [34] do not seem to match what authors showed in the original paper.
>
> ### Differences to RGCA
>
> RGCA is based on a variational Autoencoder (VAE), which learns a **personalized expression space** from dense multi-view captures of a light stage setting. The input to the encoder are the vertices of a **non-rigidly registered** template mesh, with corresponding texture maps. The decoders predict the geometric properties of Gaussian primitives along with the parameters for the precomputed radiance transfer function and specular shading.
>
> In contrast, our method builds directly on top of the **parametric** morphable model FLAME, which has a **shared expression space** across different identities. This key difference enables applications such as cross-reenactment, where expressions can be transferred between subjects, or generally speaking, an avatar can be animated with FLAME parameters from any source without the need for a personalized encoder. Therefore, we show animation from monocular videos, which is not possible with RGCA’s VAE approach.
>
> Given these FLAME parameters, we predict expression-dependent offsets for Gaussian primitives rigged to the base mesh, and a set of expression features for our appearance model, which computes diffuse reflectance with a small MLP and specular reflectance using view-dependent normals and visibility in a Cook-Torrance model. As stated in the main paper, we acknowledge that obtaining the parameters for the specular shading is inspired by RGCA [34]. We will add these clarifications in the related work section of the final paper.
>
> ### Ablation / Justification of Our Appearance Model
>
> Below, we show two additional ablations to quantify the contribution of our new appearance model. The results are extensions to the ablation study in Table 3 of the main paper and follow the same evaluation protocol. For completeness, we also evaluate novel-view synthesis on the training frames.
>
> #### NVS
>
> | Method             | PSNR  | SSIM   | LPIPS  |
> |--------------------|:-----:|:------:|:------:|
> | Ours (Full)        | 32.12 | 0.9266 | 0.0699 |
> | Spherical Gaussians | 31.86 | 0.9254 | 0.0700 |
> | PRT Diffuse        | 30.70 | 0.9031 | 0.0987 |
>
> #### NVS + Relighting
>
> | Method             | PSNR  | SSIM   | LPIPS  |
> |--------------------|:-----:|:------:|:------:|
> | Ours (Full)        | 31.38 | 0.8956 | 0.1040 |
> | Spherical Gaussians | 31.55 | 0.8953 | 0.1031 |
> | PRT Diffuse        | 29.23 | 0.8374 | 0.1577 |
>
> #### NVS + Relighting + Self-Reenactment
>
> | Method             | PSNR  | SSIM   | LPIPS  |
> |--------------------|:-----:|:------:|:------:|
> | Ours (Full)        | 31.38 | 0.8956 | 0.1040 |
> | Spherical Gaussians | 31.55 | 0.8953 | 0.1031 |
> | PRT Diffuse        | 29.23 | 0.8374 | 0.1577 |
>
> **Spherical Gaussians Specular**: In this experiment, we replace our Cook-Torrance specular term with the spherical Gaussians formulation from RGCA [34], leaving the rest of our method untouched. While we observe slightly better results in almost all metrics, we still decided in favor of the Cook-Torrance model since we observed slightly sharper specular highlights under environment relighting.
>
> **Precomputed Radiance Transfer (PRT)**: Here, we replace our neural diffuse BRDF with the PRT model from RGCA [34]. Therefore, we modify our dynamics model $F_g$ to also predict the spherical harmonics coefficients for the PRT function of the primitives, which we then integrate with the incident lighting. While we observe reasonable results under training illumination, PRT struggles to generalize to novel illuminations, which aligns with our observations from the comparisons with RGCA in Table 2 and Figures 4 and 7.
>
> We will add the qualitative comparisons of both ablations to the final version of the paper.
>
> ### Our RGCA Results Compared to the Original Evaluation
>
> RGCA results from the original paper were obtained using a capture setup with 160 cameras and 460 lights densely distributed on a 360-degree dome structure. Our comparisons were conducted with data from our capture setup, consisting of (only) 16 cameras and 40 lights, and covering only the frontal hemisphere around the head.
>
> Another difference is the evaluation protocol. While we test and visualize novel-view synthesis, unseen illuminations, and reenactment jointly in Table 2 and Figure 4, RGCA evaluates each modality separately, e.g., Figure 9 of [34] shows novel-view synthesis on a training frame and Figure 10 in [34] shows relighting on a training view and training expression which is a much less challenging setting than in our evaluation. Also, while we hold out a set of lights from training altogether, RGCA only excludes a set of light patterns, which, due to the additive nature of light, is a less meaningful test. Still, we indeed acknowledge that RGCA can produce higher-fidelity results when trained with more camera views and light patterns.
>
> To quantify these claims, we present the results of an additional evaluation on the Goliath-4 dataset, which is similar to the original training setting of RGCA. Note, however, that the publicly released dataset is subsampled to every 10th frame, and the provided images are heavily compressed. The required FLAME tracking we computed for the Goliath-4 dataset is also not perfectly accurate due to the limited time of the rebuttal period. We train both our method and RGCA in three different configurations. (1) With the full available set of cameras, holding out 10 random views for evaluation, (2) with a random subset of 16 train cameras and 4 cameras for evaluation, and (3), with a subset of 16 train and 4 test cameras sampled from the frontal region to mimic a similar setting than in our capture setup.
>
> Due to the limited time, we limit ourselves to one of the subjects (QZX685) and use the provided train/test split for evaluating unseen expressions and hold out a random subset of 10% of the total light patterns to evaluate relighting capabilities. Our method consistently outperforms RGCA on the SSIM and LPIPS metrics. We will add a qualitative comparison to the final version of the paper.
>
> #### Full Set of Cameras
>
> | Method             | PSNR  | SSIM   | LPIPS  |
> |--------------------|:-----:|:------:|:------:|
> | RGCA               | 29.89 | 0.8869 | 0.1392 |
> | Ours               | 29.70 | 0.9080 | 0.1165 |
>
> #### Random Camera Subset (10%)
>
> | Method             | PSNR  | SSIM   | LPIPS  |
> |--------------------|:-----:|:------:|:------:|
> | RGCA               | 28.74 | 0.8753 | 0.1468 |
> | Ours               | 28.68 | 0.8966 | 0.1298 |
>
> #### Frontal Camera Subset (10%)
>
> | Method             | PSNR  | SSIM   | LPIPS  |
> |--------------------|:-----:|:------:|:------:|
> | RGCA               | 29.88 | 0.8922 | 0.1326 |
> | Ours               | 29.82 | 0.9137 | 0.1139 |
>
> >Density of the lighting setup: while the setup is indeed less expensive than a full light stage, the light density seems to cause issues, in particular OLAT like reflections can be observed when the subjects are relit (see video) or when predicting albedo (Fig 5). Authors should discuss this limitation and how they plan to address it.
>
> The mentioned artifacts are also visible when training on the Goliath-4 dataset, which consists of a setup with an order of magnitude more lights than our setup. Thus, we conclude that this is a general limitation of our method, and is not caused by our dataset.
>
> The intrinsic decomposition is not perfect in the eyes region, since we use only a single specular model for all parts of the face, and eyeballs should show mirror-like reflections. Potential solutions would be an explicit eye model, such as in [Li 22, Schwartz 20], or a heterogeneous optimization/material model for the eyeballs based on the FLAME eye geometry. We will add this discussion to the final version of the paper.
>
> *Li et al. 2022 "Eyenerf: a hybrid representation for photorealistic synthesis, animation and relighting of human eyes."*
>
> *Schwartz et al. 2020 "The eyes have it: An integrated eye and face model for photorealistic facial animation."*
>
> >Albedo prediction is a simpler problem when starting with OLAT images, but in practice, this is not the actual problem to solve. Indeed a more challenging case that authors did not address is how to build an avatar from any arbitrary lighting condition, which would mimic real-world scenarios.
>
> We agree that having OLAT image data from a calibrated light stage setup simplifies the problem of recovering the geometry and material properties, and we also believe that the ultimate goal is to build relightable avatars from casually captured video data.
>
> However, there remain numerous open research questions regarding the geometry representation used to model facial expressions and hair, and also how to represent the heterogeneous materials present on human heads in a realistic and easily optimizable manner.
> We believe that finding new solutions to those open questions can legitimately be conducted in a more controlled setting, i.e., with a light stage and calibrated OLAT images, and that the generated insights will eventually also lead to research progress of relightable avatars from casually captured images. Moreover, we want to highlight our dataset contribution, which we believe will accelerate research in this direction.

---

> > ### Author Response · Authors · 2025-08-04
> > **Correction of Our Rebuttal Ablation Results**
> >
> > Dear Reviewer,
> >
> > we noticed that we accidentally posted tables with duplicate entries for our additional ablation results. We sincerely apologise for this mistake and want to correct it here:
> >
> > #### NVS
> >
> > | Method             | PSNR  | SSIM   | LPIPS  |
> > |--------------------|:-----:|:------:|:------:|
> > | Ours (Full)        | 32.12 | 0.9266 | 0.0699 |
> > | Spherical Gaussians | 31.86 | 0.9254 | 0.0700 |
> > | PRT Diffuse        | 30.70 | 0.9031 | 0.0987 |
> >
> > #### NVS + Relighting
> >
> > | Method             | PSNR  | SSIM   | LPIPS  |
> > |--------------------|:-----:|:------:|:------:|
> > | Ours (Full)        | 31.38 | 0.8956 | 0.1040 |
> > | Spherical Gaussians | 31.55 | 0.8953 | 0.1031 |
> > | PRT Diffuse        | 29.23 | 0.8374 | 0.1577 |
> >
> > #### NVS + Relighting + Self-Reenactment
> >
> > | Method             | PSNR  | SSIM   | LPIPS  |
> > |--------------------|:-----:|:------:|:------:|
> > | Ours (Full)        | 28.08 | 0.8730 | 0.1317 |
> > | Spherical Gaussians | 28.09 | 0.8729 | 0.1310 |
> > | PRT Diffuse        | 25.47 | 0.8074 | 0.1918 |

---

> > > ### Comment · Reviewer_rxBB · 2025-08-05
> > >
> > > Thank you for the additional experiments and clarifications, these address my concerns.

---

### Official Review · Reviewer_Awo4 · 2025-07-06

**Clarity:** 3
**Significance:** 4
**Originality:** 3
**Rating:** 5
**Confidence:** 4

**Summary:**

This paper proposes a method to reconstruct a relightable head avatar from a light stage, which contains multiple calibrated cameras and controllable lights. In the method section, the model learns static material properties and decodes Gaussian attributes using a CNN decoder. One of the contributions proposed by the author is to separate the shading into neural diffuse part and  analytical specular part, which can better recover the materials and obtain improved rendering results. In the experiments, compared to the previous state-of-the-art method RGCA, the proposed method can render images with more realistic lighting and finer details. Additionally, the author's claim about the release of the dataset and code is fascinating, and I believe it can further advance the development of the field of relightable head avatars.

**Questions:**

1. It seems that the quantitative results regarding the rendering speed are missing. It would be better to provide the speed data and a comparison with RGCA, since real-time rendering is claimed in the main paper. The proposed method includes many components, such as CNNs and MLPs, as well as shading computation. It would be valuable to provide the running time for each module.
2. I want to confirm whether the shadow effect is learned from the tiny neural network Fd, as mentioned in line 168 regarding "self-shadowing effects." If so, would it be possible to provide a visual result of the shadow map learned from Fd?

**Ethical Concerns:**

["NO or VERY MINOR ethics concerns only"]

**Final Justification:**

The rebuttal have well addressed my main concerns. I will keep my initial rating and lean toward acceptance for this submission.

**Limitations:**

yes.

**Quality:**

4

**Strengths And Weaknesses:**

Advantages:
1. The results of the paper and video are realistic and vivid. The proposed method can not only recover reasonable face materials from the captured data, but it is also able to render the results under challenging lighting conditions, such as OLAT light and real-world environmental lighting. Even when the avatar is driven by novel expressions and head poses, the details of the face, such as small wrinkles and hair, as well as shadows, can be correctly rendered as the environmental lighting changes.
2. The author claims to release the code and dataset, which are essential for this area. Several previous papers related to head avatar relighting captured their own datasets. However, they either do not open source their data, or the data they release is of poor quality. In this paper, the captured data includes many subjects with OLAT lights and high resolution, and the promise to release code that can reproduce the results is, in my opinion, a great contribution.
3. Although the previous method, RGCA, can render realistic relighting results, the proposed method can relight the avatar with finer details and fewer artifacts. In the experiment section, the proposed method outperforms RGCA both qualitatively and quantitatively.

Weaknesses:
1. The data capturing setup is still somewhat complicated, and the training time for a single subject is a bit long. Although these issues are well discussed in the limitations section, they are still worth mentioning, as the complicated process can prevent other users from utilizing the method. Training a prior and fine-tuning the model may help reduce the requirements for the captured data and the training time.

---

> ### Author Rebuttal · Authors · 2025-07-31
>
> >The data capturing setup is still somewhat complicated, and the training time for a single subject is a bit long
>
> Our hardware and space requirements are significantly lower compared to existing setups (see Table 1 of the main paper). We acknowledge that our method does indeed not work with casually captured data. However, we are releasing our dataset to make future research accessible to everyone.
>
> The training time (30h on a single GPU)  is comparable to concurrent and faster than previous work, which used NeRF-based representations. The primary goal of our work is to get high-quality reconstructions and run inference, i.e., relighting, animation, and novel-view synthesis, in real-time.
>
> >Training a prior and fine-tuning the model may help reduce the requirements for the captured data and the training time.
>
> Relightable Avatar reconstruction from casually captured images or videos is the ultimate goal of research in this direction. We discussed in Sec. 5.4 that learning an appearance prior of human heads is an exciting direction for future work, and we expect that the release of our dataset opens up research in this direction for a broader community.
>
> >It seems that the quantitative results regarding the rendering speed are missing. It would be better to provide the speed data and a comparison with RGCA, since real-time rendering is claimed in the main paper. The proposed method includes many components, such as CNNs and MLPs, as well as shading computation. It would be valuable to provide the running time for each module.
>
> In the table below, we summarize the runtime of the components of our method and compare it with RGCA [34]. We conducted all measurements on a single NVIDIA RTX A6000 GPU  using a UV resolution of $512^2$, which results in 202k primitives. We render images at a resolution of 1100x1604, corresponding to the training resolution of our avatars.
>
>
> | Method | CNNs | Diffuse Shading | Specular Shading | Splatting / Rasterization | Total |
> |--------|:----:|:---------------:|:----------------:|:-------------------------:|:-----:|
> | RGCA   | 9ms  | 1ms             | 1ms.             | 9ms                       | 20ms  |
> | Ours   | 4ms  | 3ms             | 1ms.             | 9ms                       | 17ms  |
>
> We will add this table to the final version of the paper. Both methods use the same gsplat splatting/rasterization backbone. Hence, we only report the rasterization speeds for completeness.
>
> >I want to confirm whether the shadow effect is learned from the tiny neural network Fd, as mentioned in line 168 regarding "self-shadowing effects." If so, would it be possible to provide a visual result of the shadow map learned from Fd?
>
> Yes, all global illumination effects (e.g., shadows, indirect illumination, etc.) are learned implicitly by the neural diffuse BSDF, since a physically-based approach would not allow for real-time
> rendering and would further increase the training time of the avatars. Since the diffuse BSDF learns all global illumination effects jointly, we cannot visualize the shadow map only. We do, however, depict the output of the diffuse BSDF in Figure 3, right to the $F_d$ network, and will add more vivid visualizations to the final paper. Further, we want to point to Figure 5, where the learned shadow can be observed in columns (c) and (e), demonstrating that it closely matches the shadow in the corresponding reference image in column (f).

---

> > ### Author Response · Authors · 2025-08-05
> >
> > Dear Reviewer Awo4,
> >
> > We appreciate your thoughtful feedback and valuable suggestions, which play a crucial role in improving the quality of our work. We hope that our rebuttal addresses your concerns, which we have summarized below:
> >
> > **Data capturing and training time**: Our setup requires significantly less hardware and space than existing methods, and we are releasing our dataset to support future research.
> >
> > **Use of priors**: Learning a prior could reduce capture and training demands. Section 5.4 discusses this as an exciting future direction, enabled by our dataset release.
> >
> > **Runtime breakdown**: We provide a detailed runtime comparison with RGCA, showing that our method achieves real-time performance.
> >
> > **Shadow effects**: Self-shadowing and other global illumination effects are learned implicitly by the neural diffuse BSDF. We intend to improve the corresponding visualizations in the final version.
> >
> > Please let us know if anything remains unclear or if you have further questions.

---

### Note · Authors · 2025-08-13

We thank all reviewers for their thoughtful and constructive feedback, which has helped us refine and strengthen our work. We are delighted that they found that BecomingLit is a “simple yet effective” (rxBB) method, that produces “very high-quality results” (B4Yt) and  “can not only recover reasonable face materials from the captured data, but it is also able to render the results under challenging lighting conditions” (Awo4). We appreciate that the reviewers share our thoughts that the release of our OLAT dataset is “a great contribution” (Awo4), “highly appreciated by the community” (CBwD), and will “accelerate the research in relightable avatars” (rxBB).

Following the initial reviews, we conducted additional experiments in our rebuttal:

**Ablation Studies**: We quantify the benefits of our hybrid shading design, demonstrating improved generalization to novel lighting situations compared to Precomputed Radiance Transfer and Spherical Gaussians.

**Goliath-4 Evaluation**: Our method outperforms RGCA on SSIM and LPIPS across full, random, and frontal camera subsets, proving our method’s performance is not tied to a single dataset.

**Runtime Breakdown**: We present real-time performance and a detailed component-wise comparison against RGCA.

Furthermore, we clarified the key methodological distinctions from RGCA, emphasizing the shared expression space that facilitates cross-subject reenactment using, for instance, monocular videos. We detailed the differences in our evaluation protocol that, in contrast to our baseline, jointly tests novel views, expressions, and lighting. This approach accounts for the observable differences in quality in our results compared to those reported in the original paper.

For the final version of the paper, we will add
- Quantitative and qualitative results from the additional appearance model ablations and the evaluation on the Goliath-4 dataset.
- Runtime details and comparisons to our baselines.
- Improved visualizations regarding the intrinsic decomposition, including normal and diffuse reflectance maps.
- A comprehensive discussion of our methods' limitations concerning the eyes and mouth interior.
- A clear description of the methodological and experimental differences from RGCA.

---

### Decision · Program_Chairs · 2025-09-17

**Decision:**

Accept (poster)

**Comment:**

This paper was reviewed by 4 experts in the field. After authors’ feedback and internal discussion, all reviewers agreed that this is solid work and should accept it (4, 4, 4, 5).

Specifically, reviewers agreed that the proposed solution of this work generates high-quality results. Particularly, the results are vivid and contain fine details. The proposed solution is interesting and novel.

Still, there are some remaining issues. For example, some ideas are from RGCA [34] and authors should further clarify that. Also, the results of [34] do not match the original results. Finally, the evaluation is limited and more results should be included. Although they should not hurt the novelty of this work, we think the authors must address them in the revision.

Considering all these, the decision is to recommend the paper for acceptance to NeurIPS 2025. We recommend the authors carefully read all reviewers’ final feedback and revise the manuscript as suggested in the final camera-ready version. We congratulate the authors on the acceptance of their paper!